# Estimates of functional muscle strength from a novel progressive lateral step-up test are feasible, reliable, and related to physical activity in children with cerebral palsy

Trevor Batson[1], Sydni V. W. Whitten[1], Harshvardhan Singh[2], Chuan Zhang[3], Gavin Colquitt [4], Christopher M. Modlesky [1] *

1 Department of Kinesiology, University of Georgia, Athens, GA, United States of America, 2 Department of Physical Therapy, University of Alabama at Birmingham, Birmingham, AL, United States of America, 3 School of Physical Education and Sports, Central China Normal University, Wuhan, China, 4 Appalachian Institute for Health and Wellness, Beaver College of Health Sciences, Appalachian State University, Boone, NC, United States of America

* christopher.modlesky@uga.edu

**Data Availability Statement:** All relevant data are within the manuscript and its Supporting Information files.

## Abstract

### Objective

To determine if estimates of functional muscle strength from a novel progressive lateral-step-up test (LSUT) are feasible, reliable, and related to physical activity in children with cerebral palsy (CP).

### Design

Cross-sectional; test-retest reliability Subjects/Patients: Children with CP and typically developing control children (n = 45/group).

### Methods

An LSUT with 10, 15, and 20 cm step heights was completed. It was repeated 4 weeks later in 20 children with CP. A composite score of LSUT was calculated based on the step height and number of repetitions completed. Physical activity was assessed using monitors worn on the ankle and hip.

### Results

Only 4 (13%) of the children with CP were unable to complete a lateral step-up repetition without assistance. All children were able to complete at least 1 repetition with assistance, though more than twice as many children with CP required assistance at 15 and 20 cm step heights than at the 10 cm step height ($p < 0.01$). Children with CP had 59 to 63% lower LSUT performance, 37% lower physical activity assessed at the ankle, and 22% lower physical activity assessed at the hip than controls (all $p < 0.01$). The intra-class correlation coefficient ranged from 0.91 to 0.96 for LSUT performance at the different step heights and was 0.97 for the composite score. All LSUT performance measures were positively related to

**Funding:** Author who received award: CMM Grant numbers: R01HD090126, R15HD071397 Full name of funder (both grants funded by same funder): Eunice Kennedy Shriver National Institute of Child Health and Human Development URL of funder website: https://www.nichd.nih.gov/ The funders had no role in study design, data collection and analysis, decision to publish, or preparation of the manuscript.

**Competing interests:** The authors have declared that no competing interests exist.

ankle physical activity in children with CP ($r$ range = 0.43 to 0.47, all $p < 0.01$). Only performance at 20 cm and the composite score were positively related to hip physical activity ($r$ = 0.33 and 0.31, respectively, both $p < 0.05$). The relationship between the LSUT performance and physical activity at both the ankle and hip increased when age and sex were statistically controlled (model $r$ range = 0.55 to 0.60, all $p < 0.001$).

## Conclusion

Estimates of functional muscle strength from a novel progressive LSUT are feasible, reliable, and positively related to physical activity in children with CP.

## Introduction

Cerebral palsy (CP) is a disorder of movement and posture resulting from a non-progressive insult to or malformation of the developing brain [1]. Children with CP have muscles that are small [2], have increased fat [2] and collagen [3] infiltration, and have poor motor unit recruitment [4], especially in the lower extremities [5], which is consistent with their substantial weakness [6]. Children with CP also have very low participation in physical activity [2, 7, 8], which contributes to their underdeveloped muscles [2, 4, 9] and bones [9–12]. Together, these deficits likely contribute to the increased risk of chronic disease observed in this population [13, 14]. Logically, muscle weakness is a key physical limitation to physical activity in children with CP, and a positive relationship has been shown between muscle strength and physical activity in other populations with physical limitations [15–17]; however, the few studies that have examined this relationship in adolescents and young adults with CP suggest it is weak [18] or undetectable [19].

It is plausible that the limited relationship between muscle strength and physical activity in children with CP is due to the methodology employed to assess muscle strength. Tests that assess functional muscle strength and involve the coordination of multiple joints to move the body through open space may be stronger predictors of physical activity than traditional muscle strength tests focused on isolated joint movements [18]. One such test is the lateral step-up test (LSUT), which has been used to assess lower-body muscle strength in individuals with CP [20–22] and involves the coordination of muscles acting at the hip, knee, and ankle [23]. Performance on the LSUT is positively associated with measures of gross motor function and mobility [20–22], and has efficacy for lower-body strengthening of children with CP [24]. Following the established core sets for children with CP as part of the International Classification of Functioning, Disability, and Health, Children and Youth (ICF-CY), the LSUT has potential to address a number of neuromusculoskeletal and movement related functions in unison (i.e. mobility and stability of lower extremity joints, muscle power and tone, and control of voluntary movement) [25]. Specifically, the LSUT relates to a child's capacity to meet challenges regarding activity and participation in daily tasks and major life areas, such as locomotion and engagement in play. However, one limitation of the LSUT employed in previous studies is the use of a single step height [20–22]. The functional task of stepping in real-world settings requires individuals with CP to adapt based on variable conditions, such as the height of a step. The addition of multiple steps adds task variability and complexity that may better transfer to real-world settings requiring individuals with CP to adapt based on variable conditions, such as the height of a step [26]. Furthermore, higher step heights may exclude younger and smaller children with CP, especially those with more extensive functional limitations. Thus, tests with

multiple step heights should enable the assessment of children with CP with a wider range of functional strength and gross motor function. Another plausible reason for the lack of a relationship observed between muscle strength and physical activity in children with CP is the methodology used to assess physical activity. Specifically, there is evidence that activity monitors worn on the ankle provide more accurate estimates of physical activity during simulated free-living conditions in children with CP than monitors worn on the hip or wrist [27].

The aims of this study were to determine: 1) if a progressive LSUT with multiple step heights vs. an LSUT with a single step height is a more feasible test to assess functional muscle strength in children with CP; 2) the inter-day reliability of performance on a progressive LSUT in children with CP; and 3) the relationship between performance on a progressive LSUT and physical activity measured at the ankle vs. the hip in children with CP. We hypothesized that a progressive LSUT provides feasible and reliable estimates of functional muscle strength that are related to physical activity in children with CP.

## Methods

### Participants and protocol

We conducted a cross-sectional study using data collected from October 1, 2012 to May 31, 2016 at the University of Delaware and from April 18, 2019 to March 4 2022 at the University of Georgia. The studies were approved by the Institutional Review Boards at each institution. The same recruitment and data collection protocols were followed at each site. Data collection at the University of Georgia is ongoing. The data collected at the University of Delaware was accessed on January 18, 2020 and authors did not have access to information that could identify the individual participants during or after data collection. Ambulatory children with unilateral and bilateral spastic CP (5–11 y) and typically developing control children similar in age and sex to children with CP were recruited to participate in the first part of the study and address the first and third aims. Children with CP who participated in the first part of the study were recruited to participate in the second part of the study to address the second aim, which included a return visit 4 weeks later.

Children were recruited from hospitals, public schools, and pediatric rehabilitation offices throughout the Southeast and Mid-Atlantic regions of the U.S. Siblings of children with CP who enrolled in the study were also recruited. Recruitment was conducted via flyers, postcards, newspaper advertisements, social media posts, and word of mouth. Exclusion criteria included prior fracture in both femurs or tibias, current use of bisphosphonates, inability to stand independently, inability to ambulate without assistance, orthopedic surgery within the last 6 months, current use of baclofen, and botulinum toxin treatment within the last year. Additional exclusion criteria for typically developing children were participation in high-level sports, outside the 5th and 95th percentiles for age- and sex-based height or body mass, a history of a neurologic disorder, motor disorder, growth disorder, or chronic disease, and chronic use (> 6 months) of medications known to affect growth. Data were collected at the University of Georgia (n = 27 children with CP and 26 controls) and the University of Delaware (n = 18 children with CP and 19 controls) and approved by their institutional review boards. All testing at both sites followed the same rigorous standardized protocol monitored by the senior author, CMM. All parents/guardians were instructed to read, and then were thoroughly verbally guided through an informed consent form outlining all known risks associated with testing before any collection of data. Written consent was then obtained from parents/guardians, while witnessed by a member of the research team. All participants received an oral explanation of the study and tests that would be performed and oral assent was obtained. Data

collected from the University of Delaware were fully anonymized before they were accessed and included in the present study.

All participants had anthropometrics, gross motor function, and the LSUT conducted during a single session. Physical activity monitors were worn within 2 weeks of the session.

## Anthropometrics

Height was assessed with a stadiometer (Seca 217; Seca GmbH & Co. KG., Hamburg, GER) to the nearest 0.1 cm. Body mass was assessed with a digital weight scale (Detecto 6550, Cardinal Scale, Webb City, MO) to the nearest 0.1 kg. Body mass index (BMI) was calculated based on height and body mass. Height, body mass, and BMI percentiles were calculated based on the Centers for Disease Control and Prevention growth charts [28]. All measurements were taken with participants in minimal clothing without shoes or bracing devices.

## Gross motor function

Gross motor function was assessed by a trained health professional using the Gross Motor Function Classification System (GMFCS), which ranges from I–V. Level I and II reflect independent ambulation, level III reflects walking with assistive devices and levels IV and V reflect wheelchair empowered mobility [29]. Children within GMFCS levels I-III participated in our study, meaning all participants with CP could ambulate, either without or with assistive devices.

## Lateral-step-up test (LSUT)

Functional muscle strength of the more affected lower limb in children with CP and the non-dominant lower limb in controls was assessed using a progressive LSUT. Participants did not wear shoes during the test. The test consisted of 3, 20-second trials, with at least 30 seconds of rest between trials. Each trial corresponded to an increase in step height (i.e., 10, 15, and 20 cm). Step heights were chosen to mimic common obstacles faced in the environment of those with CP, as 20 cm represents the maximal step height in residential areas as outlined by the International Codes Council [30] and 15 cm is the most common curb height in US streets [31]. Children with CP were instructed to stand with the more affected limb on the step and less affected limb resting on the floor, while controls were instructed to stand with their non-dominant limb on the step and dominant limb resting on the floor. Feet were parallel and approximately shoulder width apart. Participants were instructed to lift the resting limb onto the step by extending the tested limb, and then to return the resting limb to the ground by flexing the tested limb. Participants were asked to keep their hips, knees, and feet facing forward and not rotate on the platform while performing the test. They were instructed to perform as many repetitions as possible without support. If support was needed, every repetition that was not done independently (e.g., supported self by using the hand of the spotter) was noted. Steps that required assistance were adjusted by multiplying them by 0.1. The correction was based on an analysis of the children with CP who required assistance to complete a step at a particular step height. The most steps completed at 10 cm was 9, at 15 cm 8, and at 20 cm 8. By applying a correction of 0.1, 9 assisted steps would be equivalent to 0.9 steps and 8 assisted steps would be equivalent to 0.8 steps, etc. Therefore, a child would receive credit for their assisted steps, but it would not be equivalent to 1 full unassisted step. A detailed demonstration of the appropriate lateral step-up technique was performed by the administrator before the start of testing. If needed, demonstration was repeated. Practice attempts were completed by the participant before the test. A composite LSUT score was generated by multiplying repetitions at 10 cm by 1, repetitions at 15 cm by 1.5, and repetitions at 20 cm by 2, then adding the total

score. A version of the LSUT and the composite score have been used to assess functional muscle strength and its relationship with pre-frontal cortex activity in children with CP [32].

## Physical activity

Physical activity was assessed using the Actigraph GT9X (Pensacola, FL; n = 27 children with CP and n = 26 controls) or the Actical (Respironics Inc., Bend, OR; n = 18 children with CP and n = 19 controls) accelerometer-based physical activity monitors. The raw data mode used 60 second epochs to register activity counts [33]. To convert total activity counts from the Actical to the Actigraph monitor, a calibration equation was developed using data from 8 ambulatory children with CP and 9 controls (4–11 y) who wore both monitors on the same ankle for 4 days:

Ankle Actigraph total activity counts = 2.949 x Ankle Actical activity counts + 199840, $r^2$ = 0.959

Hip Actigraph total activity counts = 2.7192 x Hip Actical activity counts + 145980, $r^2$ = 0.912

Our lab previously demonstrated that the reliability of total activity counts from the Actical monitors is excellent [9].

Participants wore 2 monitors on the lateral aspect of the ankle and 2 monitors on the hip of the more affected side in children with CP and on the non-dominant side in controls. Data was collected for 4 days (3 weekdays and 1 weekend day) while participants wore monitors continuously for 24 hours. Participants and participants' families were asked to only take the monitors off during bathing, showering, or swimming. This was confirmed by reviewing activity logs kept by each participant with assistance from their parent and by visually examining the graphical output generated by the manufacturer's software. If participants did not wear the monitors on any of the days, they were asked to re-wear the monitors for missed days. Total activity counts/day averaged from the 2 monitors are reported.

## Statistical analysis

Data were analyzed using IBM SPSS Statistics (version 24, Armonk, NY). Data were checked for normality by examining skewness and kurtosis, and by conducting the Shapiro-Wilk test. Group differences in physical characteristics, LSUT performance, and physical activity counts were assessed using independent sample between-subjects t-tests if data were normally distributed, and Mann-Whitney U tests if data were not normally distributed. One sample t-tests were used to determine whether the age- and sex-based percentiles for height, body mass, and BMI percentile were different from the 50th age- and sex-based percentile in the children with CP and in the controls. The feasibility of the LSUT was determined by the number of children who could complete repetitions on at least one step with or without assistance. The McNemar test was used to determine if there were more children with CP who were able to complete at least one unassisted repetition at the 10 cm step height than at the 15 or 20 cm step heights during the LSUT. Differences in age, height, body mass, and BMI in children with CP who required assistance to complete the LSUT at a step height compared with children who required no assistance were assessed using independent t-tests if data were normally distributed, and Mann-Whitney U tests if the data were non-normally distributed. Group differences in LSUT repetitions at each step height were assessed using 2-way between-subjects ANOVA with repeated measures and Bonferroni post hoc tests. Alpha level was set at 0.05. All tests were 2-tailed. Effect size was determined using Cohen's $d$ ($d$), with 0.2, 0.5, and 0.8 representing small, medium, and large effect sizes, respectively. Bivariate linear regression analysis was used to determine relationships between LSUT performance and physical activity in each

group independently. Multiple linear regression analysis was used to determine the amount of variance in physical activity in children with CP explained by LSUT repetitions at each step height (or LSUT composite score), age, sex, and CP type.

The within-rater test-retest reliability of LSUT performance at each step height and the composite score was assessed in children with CP using the ICC with 95% confidence intervals for a 2-way random effects model and absolute agreement. An ICC < 0.50 indicates poor reliability, between 0.50 and 0.75 indicates moderate reliability, between 0.75 and 0.90 indicates good reliability and > 0.90 indicates excellent reliability [34]. Agreement between performance on the first and second LSUT at each step height and the LSUT composite score was assessed using the standard error of measurement (SEM), which was calculated as the square root of the variance. The minimal (or smallest) detectable change (MDC) represents the smallest change that can be detected by performance on an individual step height during the LSUT or overall performance as determined using the LSUT composite score due to growth, intervention, injury, sickness, etc. The MDC was calculated using the following formula, $1.96 \times \sqrt{2} \times SEM$, as explained by de Vet et al. [35]. The minimal important difference (MID) represents the smallest difference that patients view as enough of a change to justify modification of their management in the absence of troublesome side effects and excessive cost [36]. The MID was calculated using the following formula [37, 38]:

$$MID = X \times SD \text{ of Test } 1 \times \sqrt{(1-r)}$$

X = 2.77 if it is anticipated that a large effect is needed and 1.96 if it is anticipated that a medium effect is needed. Agreement was also evaluated using a one-way within-subjects ANOVA with repeated measures and Bonferroni corrections, linear regression analysis, and Bland-Altman plots [39].

## Results

### Physical characteristics and LSUT performance

Forty-five children with CP and 45 typically developing control children matched to children with CP for age and sex participated in the cross-sectional study. Physical characteristics are summarized in Table 1. Children with CP had lower height percentile compared to controls

**Table 1. Characteristics of children with cerebral palsy (CP) and typically developing children (Con).**

| Variable | CP (n = 45) | Con (n = 45) | d | p |
|---|---|---|---|---|
| Age (y) | 8.5 (2.2) | 8.2 (2.2) | 0.013 | 0.952 |
| Sex (M/F) | 27/18 | 27/18 | 0.000 | 1.000 |
| Height (m) | 1.25 (0.14) | 1.29 (0.13) | 0.298 | 0.161 |
| Height (%) | 33 (30) | 57 (30) | 0.772 | <0.001 |
| Body mass (kg) | 27.1 (8.7) | 27.9 (8.4) | 0.099 | 0.638 |
| Body mass (%) | 42 (33) | 53 (29) | 0.363 | 0.088 |
| BMI (kg/m$^2$) | 16.9 (3.4) | 16.4 (2.4) | 0.187 | 0.377 |
| BMI (%) | 50 (35) | 49 (29) | 0.058 | 0.783 |
| GMFCS (I/II/III) | 31/12/2 | | | |
| CP type (unilateral/bilateral) | 23/22 | | | |
| MAS$_{MAL}$ (0/1/1.5/2/3) | 2/9/17/9/8 | | | |
| MAS$_{LAL}$ (0/1/1.5/2/3) | 7/18/9/7/4 | - | | |

Values are mean (SD); % reflects the percentile relative to age- and sex-based norms; BMI = body mass index; GMFCS = Gross Motor Function Classification System; MAS$_{MAL}$ = Modified Ashworth Scale on the more affected limb; MAS$_{LAL}$ = Modified Ashworth Scale on the less affected limb.

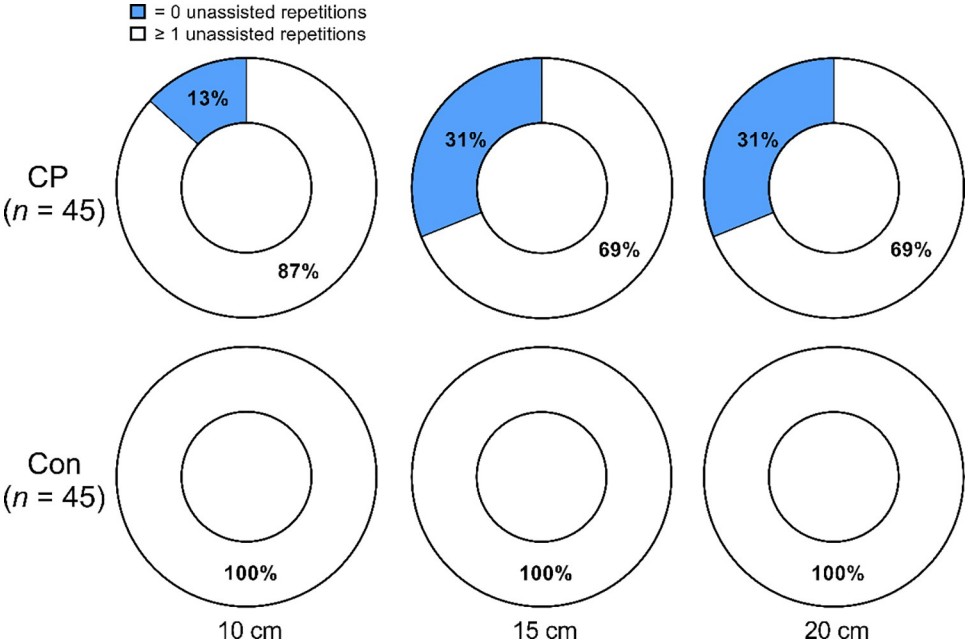

**Fig 1. Percentage of children able to complete at least one unassisted repetition at each step height of the lateral step-up test (LSUT).** Ring charts demonstrate the percentage of children with cerebral palsy (CP) and typically developing control children (Con) who were able to complete one or more repetitions at 10 cm, 15 cm, and 20 cm step heights of the LSUT.

($p < 0.001$). For children with CP, height percentile ($p < 0.001$) and body mass percentile ($p = 0.044$) were significantly lower from the age- and sex-based 50th percentiles. Height, body mass, and BMI percentiles of controls were not different from the age- and sex-based 50th percentile ($p > 0.140$). LSUT performance at each step height and the composite score were positively related to age in children with CP ($r$ range = 0.669 to 0.732, all $p < 0.001$).

All controls were able to complete repetitions unassisted at all step heights in the LSUT. All children with CP were able to complete repetitions at one or more step heights in the LSUT. However, some children with CP required assistance, even at the lowest step height, as demonstrated in **Fig 1**. More than twice as many children required assistance at the 15 cm ($p = 0.008$) and 20 cm ($p = 0.008$) step heights than at the 10 cm step height. The children who required assistance to complete the 15 and 20 cm step heights (n = 14) were younger and shorter than children who required no assistance (n = 31; all $p < 0.05$).

Bar graphs demonstrating group differences in LSUT performance are in **Fig 2**. Compared to controls, children with CP performed 59 to 63% fewer repetitions at each LSUT step height ($d$ range = 2.064 to 2.367, all $p < 0.001$) and had a 62% lower LSUT composite score ($d = 2.234$, $p < 0.001$). There were no sex differences in LSUT performance in either group ($p > 0.05$).

## Test-retest analysis

Twenty children with CP who completed the first part of the study returned 4 weeks later and completed a second progressive LSUT. A summary of the test-retest LSUT data are in **Table 2**. There was no significant inter-day difference in the LSUT repetitions at any step height or the composite score ($p > 0.05$). The ICC's ranging from 0.91 and 0.96 and the ICC 95% confidence intervals ranging from 0.80 to 0.99 for the LSUT repetitions at the individual step

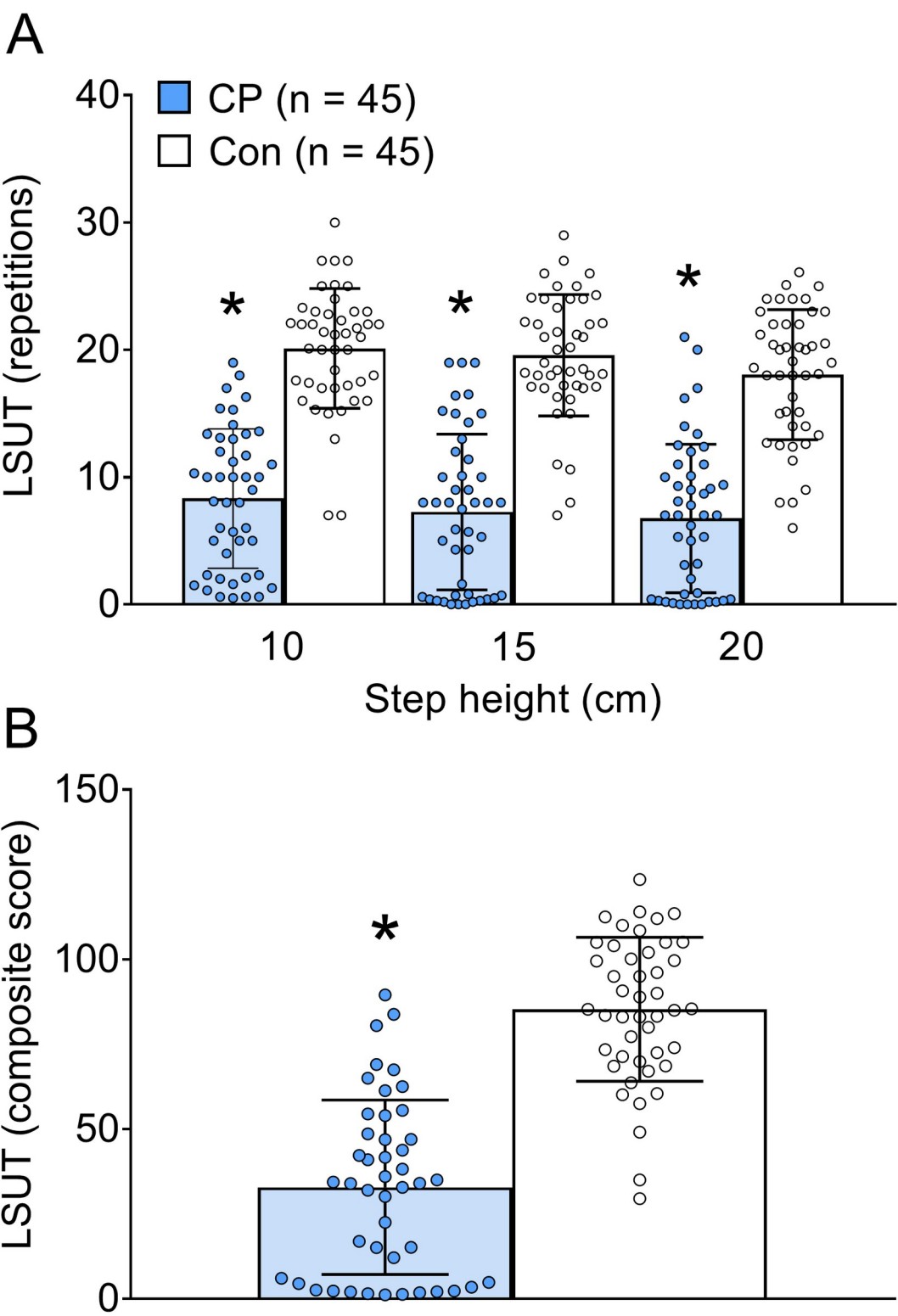

**Fig 2. Group differences in lateral step-up test performance (LSUT).** Bar graphs demonstrate group differences in: A) number of repetitions completed at the 10 cm, 15 cm, and 20 cm step heights of the LSUT, and B) the composite score of LSUT performance in children with cerebral palsy (CP) and typically developing control children (Con). Values are means ± SE. *Different from controls, $p < 0.05$.

**Table 2. Inter-day, test-retest lateral-step-up test (LSUT) repetitions at different step heights and composite score in children with cerebral palsy (n = 20).**

| Measure | Test 1 repetitions | Test 2 repetitions | ICC (95%CI) | SEM | SEM % | MDC | MID$_{2.77}$ | MID$_{1.96}$ |
|---|---|---|---|---|---|---|---|---|
| LSUT | | | | | | | | |
| 10 cm | 7.5 (5.8) | 7.9 (6.9) | .91 (.80, .96) | 1.9 | 24 | 5.2 | 4.4 | 3.1 |
| 15 cm | 7.1 (6.4) | 7.5 (7.2) | .96 (.91, .99) | 1.3 | 18 | 3.6 | 3.1 | 2.2 |
| 20 cm | 7.5 (5.8) | 7.8 (6.9) | .95 (.87, .98) | 1.4 | 20 | 3.8 | 3.5 | 2.6 |
| Composite | 31.3 (27.1) | 33.5 (29.9) | .97 (.94, .99) | 4.6 | 12 | 12.7 | 10.3 | 7.3 |

Test 1 and Test 2 values are means (SD). ICC = intraclass correlation coefficient; SEM = standard error of measurement; SEM % = (SEM / Test 1 repetitions) x 100; MDC = minimum detectable change; MID$_{2.77}$ = Minimal important difference assuming a large effect; MID$_{1.96}$ = Minimal important difference assuming a medium effect.

heights (all $p < 0.001$) indicated good to excellent reliability. The SEM ranged from 1.3 to 1.9 repetitions and the relative SEM ranged from 18 to 24% at each LSUT step height. The ICC and ICC 95% confidence intervals were greater and the relative SEM (SEM %) was lower for the LSUT composite score than LSUT repetitions at each step height. Scatter plots (**Fig 3A– 3D**) and Bland-Altman plots (**Fig 3E–3H**) suggest that there was no bias in the test-retest LSUT measurements and they were consistent across the different levels of performance (i.e., repetitions and composite score). However, there was somewhat better reliability and agreement with the LSUT composite score than performance at any individual step height.

## Physical activity and its relationship with LSUT performance

Bar graphs demonstrating group differences in physical activity are in **Fig 4**. Children with CP had 37% lower physical activity counts at the ankle ($d = 1.378$, $p < 0.001$) and 22% lower total physical activity counts at the hip ($d = 0.745$, $p = 0.003$). Hip physical activity was lower in girls with CP than boys with CP ($d = 0.626$, $p = 0.048$). Scatter plots demonstrating the relationship between LSUT performance and physical activity in the total sample are in **Fig 5**. Number of repetitions performed at each LSUT step height and the LSUT composite score were positively related to ankle physical activity counts in children with CP ($r$ range = 0.43 to 0.47, all $p < 0.01$), but weakly and not significantly related in typically developing children ($r$ range = 0.09 to 0.14, all $p > 0.250$). Number of repetitions performed at 20 cm and the LSUT composite score were significantly related to hip physical activity counts in children with CP ($r = 0.33$ and 0.31, respectively, both $p < 0.05$), but the relationship between repetitions at 10 and 15 cm did not reach statistical significance ($r = 0.26$ and 0.29, respectively, $p = 0.076$ and 0.056). There was no significant relationship between LSUT performance and hip physical activity in controls ($r$ range = 0.00 to 0.13, all $p > 0.350$). All relationships between LSUT performance and ankle physical activity remained significant when type of CP was controlled for (all $p < 0.01$), but the relationships between LSUT performance and hip physical activity were no longer significant ($p > 0.05$).

Results from the multiple linear regression analysis in children with CP are summarized in **Table 3**. When age and sex were included in a regression model with LSUT repetitions at a step height or the LSUT composite score, the prediction of ankle physical activity counts was greater than the model that only included LSUT. However, in each model, the LSUT outcome (i.e., repetitions at a given step height or LSUT composite score) was the strongest predictor as indicated by its higher standardized β than age or sex. Age was a significant negative predictor of ankle physical activity counts in all the models, indicating that physical activity decreased with age. Sex did not have a statistically significant effect in any model ($p > 0.100$).

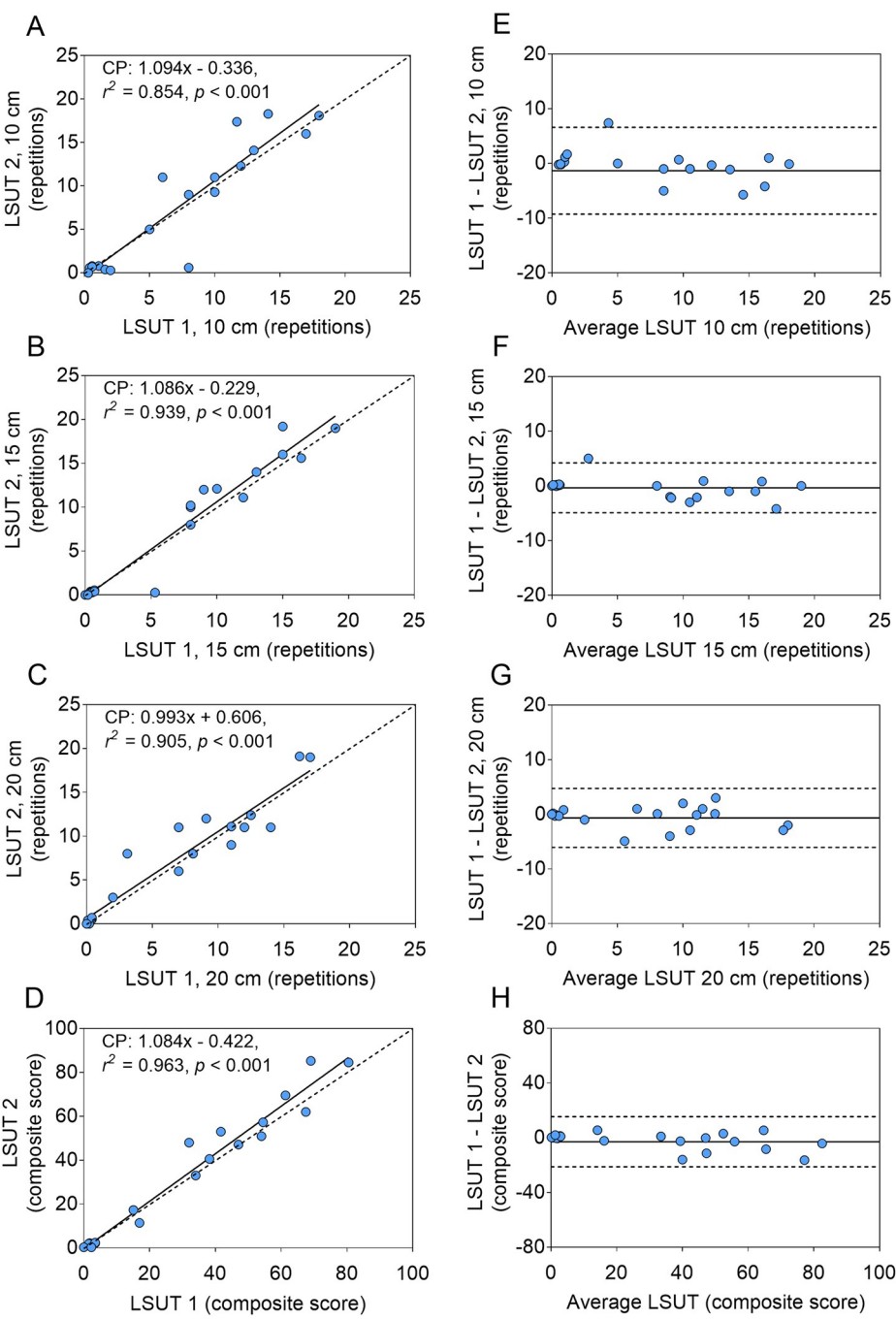

**Fig 3. Test-retest reliability and agreement in lateral step-up test (LSUT) performance.** Scatter plots (A-D) and Bland-Altman plots (E-H) demonstrate the test-retest reliability and agreement, respectively, of repetitions completed at each step height of a LSUT at 10 cm, 15 cm, and 20 cm and the LSUT composite score in children with cerebral palsy (CP).

When age and sex were included in a regression model with LSUT repetitions at a step height or the LSUT composite score, the prediction of hip physical activity counts was significant (all $p < 0.01$). Moreover, in each model, the LSUT outcome (i.e., repetitions at a given step height or LSUT composite score) was a positive predictor of hip physical activity counts

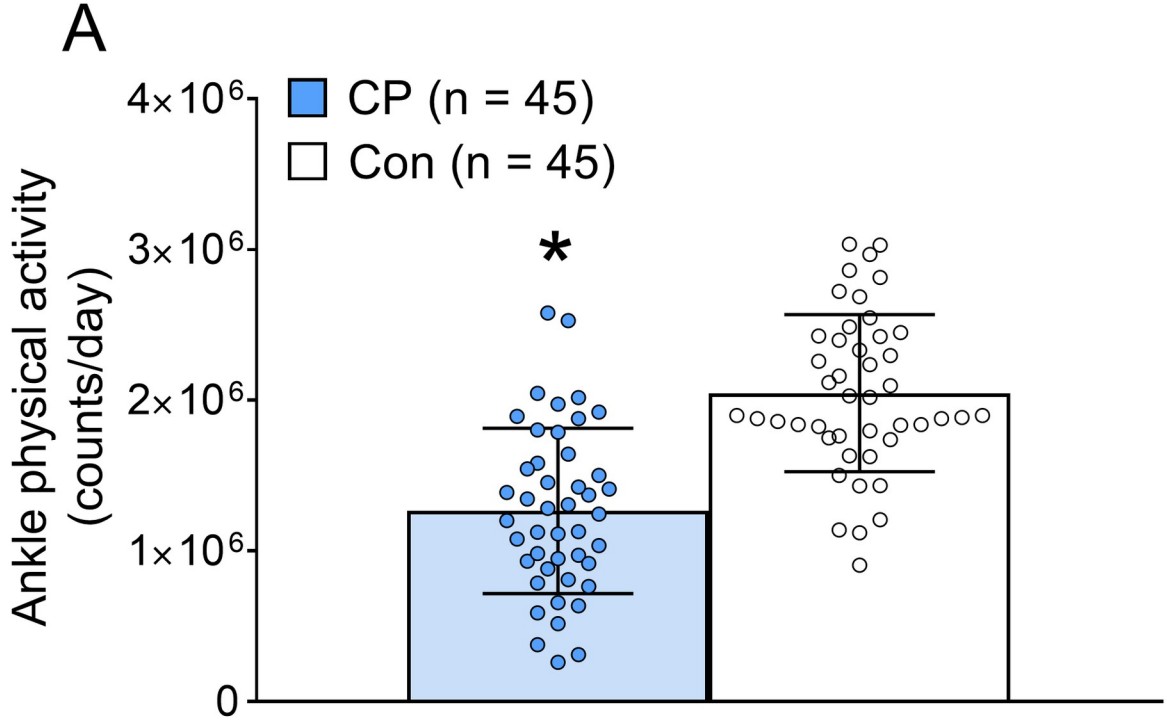

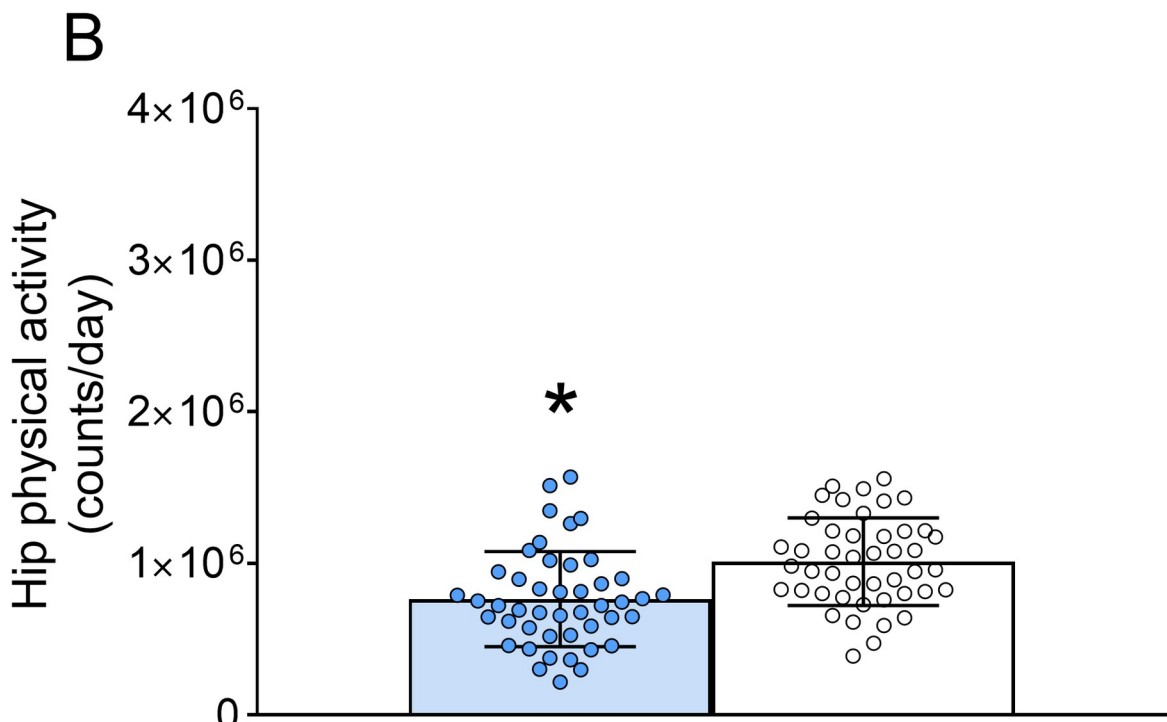

**Fig 4. Group differences in physical activity.** Bar graphs demonstrate group differences in A) ankle and B) hip physical activity counts in children with cerebral palsy (CP) and typically developing control children (Con). Values are means ± SE. *Different from controls, $p < 0.05$.

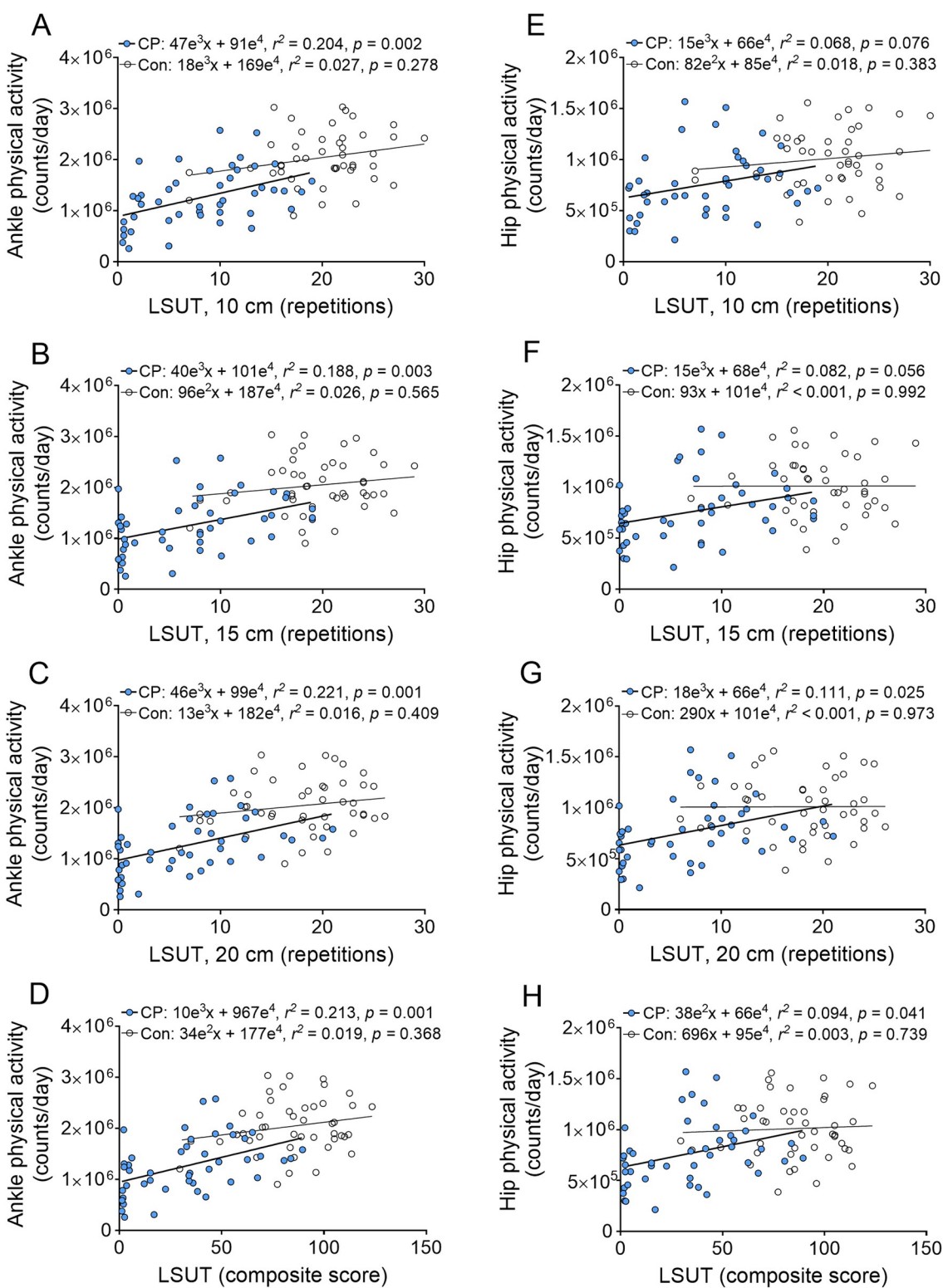

**Fig 5. Relationship between performance on the lateral step-up test (LSUT) and physical activity.** Scatter plots demonstrate the relationship between (A-D) ankle and (E-H) hip physical activity counts and repetitions completed at 10 cm, 15 cm, and 20 cm step heights of the LSUT and the LSUT composite score in children with cerebral palsy (CP) and typically developing control children (Con).

**Table 3. Multiple linear regression predicting physical activity counts at the ankle and hip in children with cerebral palsy using lateral step-up (LSUT) performance, age, and sex.**

| Measure | Coefficients | β | t-value | SE | p | Std β | 95% CI | Model R², adj R² |
|---|---|---|---|---|---|---|---|---|
| Ankle | | | | | | | | 0.317, 0.249* |
| (cts/day) | | | | | | | | |
| | Intercept | 1748032 | 4.833 | 361705 | <0.001 | | 1016999 to 2479065 | |
| | LSUT 10 cm | 58820 | 3.479 | 16907 | <0.001 | 0.583 | 24650 to 92991 | |
| | Age | -87757 | -2.106 | 41668 | 0.042 | -0.337 | -171970 to -3543 | |
| | Sex | -237743 | -1.541 | 154297 | 0.131 | -0.206 | -549589 to 74102 | |
| | CP type | -69344 | -0.418 | 165756 | 0.678 | -0.061 | 404349 to 265662 | |
| | | | | | | | | 0.307, 0.237* |
| | Intercept | 1816665 | 5.009 | 362653 | <0.001 | | 1083715 to 2549615 | |
| | LSUT 15 cm | 53198 | 3.365 | 15811 | 0.002 | 0.571 | 21242 to 85154 | |
| | Age | -85987 | -2.045 | 42037 | 0.047 | -0.330 | -170948 to -1026 | |
| | Sex | -213567 | -1.381 | 154664 | 0.175 | -0.185 | -526155 to 99021 | |
| | CP type | -68665 | -0.409 | 167956 | 0.685 | -0.061 | -408117 to 270787 | |
| | | | | | | | | 0.341, 0.275* |
| | Intercept | 1814170 | 5.133 | 353440 | <0.001 | | 1099841 to 2528499 | |
| | LSUT 20 cm | 59970 | 3.743 | 16024 | <0.001 | 0.612 | 27585 to 92355 | |
| | Age | -90803 | -2.231 | 40693 | 0.031 | -0.349 | -173047 to -8560 | |
| | Sex | -177021 | -1.180 | 150057 | 0.245 | -0.154 | -480298 to 126255 | |
| | CP type | -62787 | -0.388 | 161915 | 0.700 | -0.056 | -390030 to 264455 | |
| | | | | | | | | 0.337, 0.271* |
| | Intercept | 1793231 | 5.052 | 354944 | <0.001 | | 1075863 to 2510599 | |
| | LSUT$_{composite}$ | 13576 | 3.698 | 3671 | <0.001 | 0.614 | 6157 to 20996 | |
| | Age | -92952 | -2.253 | 41257 | 0.030 | -0.357 | -176335 to -9570 | |
| | Sex | -209500 | -1.387 | 151025 | 0.173 | -0.182 | -514734 to 95734 | |
| | CP type | -54604 | -0.334 | 163639 | -0.334 | -0.048 | -385330 to 276122 | |
| Hip | | | | | | | | 0.352, 0.287† |
| (cts/day) | | | | | | | | |
| | Intercept | 1349885 | 6.840 | 197358 | <0.001 | | 951010 to 1748760 | |
| | LSUT 10 cm | 22772 | 2.469 | 9224 | 0.018 | 0.403 | 4127 to 41417 | |
| | Age | -58644 | -2.579 | 22735 | 0.014 | -0.402 | -104594 to -12694 | |
| | Sex | -210925 | -2.505 | 84189 | 0.016 | -0.327 | -381078 to -40772 | |
| | CP type | -122207 | -1.351 | 90442 | 0.184 | -0.193 | -290063 to -60583 | |
| | | | | | | | | 0.358, 0.294† |
| | Intercept | 1374600 | 7.034 | 195429 | <0.001 | | 979624 to 1769577 | |
| | LSUT 15 cm | 21797 | 2.558 | 8521 | 0.014 | 0.417 | 4576 to 39018 | |
| | Age | -59760 | -2.638 | 22653 | 0.012 | -0.409 | -105544 to -13976 | |
| | Sex | -2102753 | -2.433 | 83346 | 0.020 | -0.314 | -371203 to -34304 | |
| | CP type | -116227 | -1.284 | 90509 | 0.206 | -0.184 | -299153 to 66699 | |
| | | | | | | | | 0.388, 0.327† |
| | Intercept | 1372032 | 7.191 | 190811 | <0.001 | | 986390 to 1757675 | |
| | LSUT 20 cm | 25666 | 2.967 | 8651 | 0.005 | 0.468 | 8183 to 43150 | |
| | Age | -63277 | -2.880 | 21969 | 0.006 | -0.434 | -107678 to -18876 | |
| | Sex | -188072 | -2.322 | 81011 | 0.025 | -0.292 | -351801 to -24344 | |
| | CP type | -109089 | -1.248 | 87413 | 0.219 | -0.173 | -285757 to 67578 | |
| | | | | | | | | 0.375, 0.313† |
| | Intercept | 1364895 | 7.071 | 193029 | <0.001 | | 974768 to 1755022 | |

(*Continued*)

**Table 3.** (Continued)

| Measure | Coefficients | β | t-value | SE | p | Std β | 95% CI | Model $R^2$, adj $R^2$ |
|---------|-------------|-----|---------|-----|-----|-------|--------|-----------------------|
| | $LSUT_{composite}$ | 5576 | 2.793 | 1996 | 0.008 | 0.450 | 1541 to 9612 | |
| | Age | -62698 | -2.794 | 22437 | 0.008 | -0.430 | -108044 to -17352 | |
| | Sex | -201134 | -2.449 | 82132 | 0.019 | -0.312 | -367130 to -35139 | |
| | CP type | -110204 | -1.238 | 88992 | 0.223 | -0.174 | -290063 to -69655 | |

Physical activity in counts/day (cts/day); LSUT 10 cm, 15 cm, and 20 cm = repetitions at 10, 15, and 20 cm step heights, respectively; Sex: 0 = male, 1 = female; CP type: 1 = unilateral, 2 = bilateral

*$p < 0.05$

†$p < 0.001$.

(all $p < 0.001$). The LSUT outcome was also the strongest predictor in each of the models. Age was a significant negative predictor of both ankle and hip physical activity counts in all models (all $p < 0.01$), indicating that physical activity counts decreased with age. Sex was also a negative predictor of hip physical activity counts in all models ($p < 0.05$), indicating that hip physical activity counts in children with CP were lower in girls than boys.

## Discussion

The present study demonstrates that a progressive LSUT including multiple step heights enables the evaluation of lower-body functional muscle strength in more children with CP than an LSUT that includes a single step height. The present study is also the first to report inter-day, test-retest reliability and agreement of performance on a progressive LSUT in children with CP. High ICC's and ICC 95% confidence intervals ($> 0.75$) indicate that LSUT performance has good to excellent inter-day reliability in children with CP. Moreover, the present study is the first to demonstrate a positive relationship between lower-body functional muscle strength, as assessed using a novel progressive LSUT, and physical activity in ambulatory children with CP. This unique observation suggests that lower-body functional muscle strength is an important factor in the physical activity of ambulatory children with CP and the methods used to assess them should be considered when evaluating their relationship. Its robust relationship with physical activity suggests that performance on the LSUT relates to the ICF-CY core sets regarding mobility and stability of lower extremity joints, muscle power and tone, and control of voluntary movement in a way that is reflective of a child's capacity to meet challenges relating to activity and participation in daily activities such as locomotion and engagement in play. Together, the observations from the present study are clinically relevant and highlight a promising method to assess functional muscle strength and the effectiveness of targeted interventions.

Prior studies have used the LSUT to assess functional muscle strength in individuals with CP. However, most of these studies were in adolescents [20, 21], a mix of ages with an unknown proportion of children [40] or adults [22], and, to our knowledge, no studies used an LSUT with multiple step heights. The importance of using multiple step heights to assess functional muscle strength was demonstrated by the observation that only 69% of the children could complete any unassisted repetitions at the 15 or 20 cm step heights, which are in the range of typical step heights used as part of an LSUT [20–22]. On the other hand, 87% were able to complete the LSUT at the 10 cm step height without assistance. Thus, including a step height of 10 cm as part of the LSUT allowed for an evaluation that spanned a wider range of functional muscle strength. The number of children able to participate in the LSUT was

increased further by providing them with assistance when needed and adjusting the repetitions and overall score. The LSUT performance was positively related to age in children with CP. Furthermore, children with CP who required assistance to complete the LSUT at 15 and 20 cm were younger and shorter than the children who required no assistance. These observations suggest that age-specific norms for the LSUT should be developed to assess functional muscle strength in children with CP. Norms based on height, knee height, and/or tibia length may also improve the assessment of functional muscle strength in children with CP using the LSUT.

Further support for the adoption of an LSUT with multiple steps rather than a single step is the success of rehabilitation approaches for those with neurological disorders that incorporate motor learning paradigms, such as practice variability [26]. The multiple step heights LSUT in the current study incorporate an element of variability in both complexity and intensity that would further support the transfer of performance on the LSUT to capacity to participate in daily activities and major life areas [41]. Therefore, in addition to having utility as an assessment of functional muscle strength, the progressive LSUT is likely to have value as part of targeted interventions aiming to address core sets related to mobility and function of the lower extremities.

Reliability was stronger for LSUT performance estimates of lower body functional muscle strength when multiple step heights were used than any single step height (ICC = 0.97); however, even LSUT performance at individual step heights showed good to excellent reliability with ICC ranging from 0.91 to 0.96 (95% ICC range = 0.80 to 0.99). The high degree of reliability is consistent with the reliability reported in a previous study of older children with CP (7 to 17 y) [40] and greater than reported in a study of similarly aged children with CP (4 to 10 y) [42], both of which included a single step height and a longer duration test (30 vs 20 s per step height in the present study). The agreement of the test, as reflected by the SEM, was stronger in the present study ranging from 1.3 to 1.9 repetitions, than in the previous studies (2.5 and 3 repetitions). The MDC, which represents the minimum change in a test or score needed to be considered an improvement or loss, was lower (range = 3.6 to 5.2 repetitions) than the MDC calculated using the SEM data from the previous study of older children with CP [40] and lower than the MDC reported for the study of similarly-aged children with CP [42], which ranged from 6.7 to 9.3 repetitions. As with ICC, the relative SEM (SEM %) was lowest for the LSUT when multiple steps were used, suggesting that the graded assessment of functional muscle strength has better agreement than assessment using a single step.

To assess the relative change needed to be considered an improvement or loss, we compared the MDC in the present study to the SD for repetitions at each step height. The change needed to demonstrate improvement or loss is largest for the 10 cm step height (0.89 SD), intermediate for the 15 cm (0.56 SD) and 20 cm (0.63 SD) step heights, and smallest for the composite score (0.47 SD). For comparison, the change needed to demonstrate improvement in the previous study of older children with CP was 0.64 and 0.69 SD [40], and in the previous study of similarly aged children with CP was 0.88 and 1.12 SD [42] for the 2 lower limbs. We also reported the MID, which is an estimate of the smallest difference that patients view as enough of a change to justify modification of their management. When we assume a large effect would be required, the change needed was largest for the 10 cm step height (0.76 SD), intermediate for the 15 cm (0.52 SD) and 20 cm (0.61 SD) step heights, and smallest for the composite score (0.38 SD). If we assume a moderate effect would be required, the change needed was even smaller, ranging from 0.27 to 0.54 SD. Hence, when considering the degree of the difference in LSUT performance between children with CP and controls in the present study ($d > 2$ SD), the estimated changes needed to capture a real change (~0.5 to 0.9 SD) and a change that is meaningful enough to warrant modifying the clinical management of a child with CP (~0.4 to 0.8 SD) seem reasonable, especially for the composite score.

In the present study, the stronger relationship between lower-body functional muscle strength estimates from the LSUT and physical activity in children with CP when physical activity was assessed at the ankle rather than the hip demonstrates the site-specificity of physical activity assessment using accelerometer-based monitors. This is consistent with prior evidence that monitors worn on the ankle provide more accurate estimates of physical activity during simulated free-living conditions in children with CP than monitors worn on the hip or wrist [27]. It is plausible that the ankle monitors are more sensitive to joint movement due to their distal location and the lower amount of soft tissue below the ankle versus the hip. Ground reaction forces may be attenuated at the hip during activities that require high lower-body functional muscle strength of children with CP. This hypothesis is consistent with the observation that the difference in physical activity between children with CP and controls was larger when assessed at the ankle (37% lower; $d$ = 1.378) than at the hip (22% lower; $d$ = 0.745). Assessment of physical activity at the hip may be one reason a previous study did not detect a relationship between a lower-body functional muscle strength and physical activity in children and adolescents with CP [19]. However, it is important to note that, in the present study, the relationship between lower-body functional muscle strength estimates from LSUT and hip physical activity improved when age and sex were included in the regression model.

In children with CP, LSUT performance was the strongest predictor of physical activity assessed at the ankle and the hip in our regression models. The standardized β value suggests that a 25.7 point increase in LSUT composite score (i.e., 1 SD) would correspond with a 365,685 (i.e., 0.641 SD) increase in daily activity counts at the ankle. This is a 28% increase in daily activity for the average child with CP in this sample. Similarly, a 5 to 6 repetition increase (i.e., 1 SD) at 10, 15, or 20 cm step heights would correspond to a 26 to 28% increase in daily activity for the average child with CP in this sample. The inclusion of age or, to a lesser extent, height improved the strength of the models presented in predicting physical activity assessed at both the ankle and hip when compared to models including LSUT performance alone. This is further evidence that age-specific (and/or size-specific) norms would improve the ability of the progressive LSUT to assess lower-body functional muscle strength in children with CP. The inclusion of sex further improved the model that predicted physical activity at the hip, and it indicted that hip physical activity was lower in girls than boys and sex should be considered when evaluating physical activity in children with CP and how it relates to functional strength.

The present study has notable strengths. The sample size was reasonably large for a study involving children with CP, and the level of CP and age range were reasonably narrow. Furthermore, controls were matched to children with CP for age and sex, were the same composition of race, and were not different from the 50[th] age- and sex-based percentiles for height, body mass, and BMI. Therefore, we are confident that the observations are applicable to children with CP who can ambulate independently and are between 5 and 11 years of age. Another strength of the study is that a scaled response to a novel progressive LSUT extending across 3 increasing step heights was captured in children with CP. Furthermore, a direct comparison of the feasibility and reliability of lower body functional muscle estimates from a single step and multiple steps was conducted.

The present study is not without limitations. The LSUT did not account for participant age or size. LSUT performance was positively related to age and height in children with CP and controls. Furthermore, children with CP who were unable to complete the 15 and 20 cm step heights of the LSUT were younger and shorter. If the progressive LSUT developed in the present study is to be used as a clinical test in the future, establishing age- and/or height-based norms may be necessary. Additionally, some children with CP were unable to independently complete the LSUT at any height. However, these children were provided with assistance and the scores were adjusted to reflect their lower functional strength. Future studies may explore

different methods for adjusting scores based on level of assistance, or the addition of a lower step height or lateral stepping without a step to include children at lower functional levels. Lastly, the cross-sectional design of the study limits the interpretation of the results. Longitudinal studies are needed to determine the influence of functional muscle strength on physical activity participation in children with CP. There may be concern about the sample size used to determine the reliability on the LSUT in the present study (n = 20). A recent study used simulated data to create an algorithm to determine appropriate sample sizes for reliability studies [43]. Minimum sample sizes for reliability studies with expected ICCs of 0.60, 0.70, and 0.80 and a default confidence interval width of 0.30 are 100, 50, and 30, respectively. Although the algorithm can't estimate a sample size for tests with an expected ICC > 0.90, which was the ICC in the present study, a sample size of 20 seems reasonable. Furthermore, to alleviate unnecessary burden on research participants, it has been recommended that researchers minimize the amount of testing conducted to estimate ICC and SEM [43]. Moreover, a study by Walter et al. [44] indicates that 20 participants is sufficient to assess reliability using 2 measurements if the ICC is high. We also acknowledge the possible variability introduced by including data collected from 2 different testing sites that extended across a 10-year period. However, all testing followed a rigorous standardized protocol monitored by the senior author, CMM. While the MDC reported for each step height and the composite score are similar to those reported in previous studies involving the LSUT in children with CP [40, 42], the LSUT may not be sensitive enough to detect small changes in functional strength to measure efficacy of short, low-intensity interventions [45]. Lastly, there is evidence that the distribution-based approach used to estimate MID in the present study does not consistently correlate with the anchor-based MID [38], and the utilization of both distribution- and anchor-based MID is regarded as the gold standard [46]. However, it has been suggested that MID estimates from the distribution-based approach be used until anchor-based estimates are determined [38].

In conclusion, the results of the present study suggest that functional-muscle-strength estimates from a novel progressive LSUT are feasible and reliable in children with CP. Furthermore, lower-body functional muscle strength estimated using a progressive LSUT is associated with low participation in physical activity in children with CP. The high degree of inclusivity and convenience of the protocol, the strong inter-day reliability, and the strength of the relationship with physical activity suggest that the progressive LSUT developed in the present study can be a useful clinical tool to assess lower-body functional muscle strength and the effectiveness of interventions in ambulatory children with CP.

## Supporting information

**S1 Table. Linear regression predicting physical activity counts at the ankle and hip in children with cerebral palsy using lateral step-up (LSUT) performance.**
(DOCX)

**S2 Table. Linear regression predicting physical activity counts at the ankle and hip in typically developing control children using lateral step-up (LSUT) performance.**
(DOCX)

## Acknowledgments

We thank all participants and their families.

## Author Contributions

**Conceptualization:** Harshvardhan Singh, Christopher M. Modlesky.

**Data curation:** Trevor Batson, Sydni V. W. Whitten, Christopher M. Modlesky.

**Formal analysis:** Trevor Batson, Christopher M. Modlesky.

**Funding acquisition:** Christopher M. Modlesky.

**Investigation:** Trevor Batson, Sydni V. W. Whitten, Harshvardhan Singh, Chuan Zhang, Gavin Colquitt, Christopher M. Modlesky.

**Methodology:** Harshvardhan Singh, Christopher M. Modlesky.

**Project administration:** Sydni V. W. Whitten, Christopher M. Modlesky.

**Resources:** Christopher M. Modlesky.

**Supervision:** Christopher M. Modlesky.

**Validation:** Trevor Batson, Christopher M. Modlesky.

**Visualization:** Trevor Batson, Christopher M. Modlesky.

**Writing – original draft:** Trevor Batson.

**Writing – review & editing:** Trevor Batson, Sydni V. W. Whitten, Harshvardhan Singh, Chuan Zhang, Gavin Colquitt, Christopher M. Modlesky.

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
