## [Decision Letter · Decision Letter 0]

15 Dec 2023

PONE-D-23-23848Estimates of functional muscle strength from a novel progressive lateral step-up test are feasible, reliable, and related to physical activity in children with cerebral palsyPLOS ONE

Dear Dr. Modlesky,

Thank you for submitting your manuscript to PLOS ONE. After careful consideration, we feel that it has merit but does not fully meet PLOS ONE’s publication criteria as it currently stands. Therefore, we invite you to submit a revised version of the manuscript that addresses the points raised during the review process.

We look forward to receiving your revised manuscript.

Kind regards,

Renato S. Melo, PhD

Academic Editor

PLOS ONE

Journal Requirements:

Reviewers' comments:

Reviewer's Responses to Questions

**Comments to the Author**

1. Is the manuscript technically sound, and do the data support the conclusions?

Reviewer #1: Yes

Reviewer #2: Partly

Reviewer #3: Yes

2. Has the statistical analysis been performed appropriately and rigorously? 

Reviewer #1: Yes

Reviewer #2: Yes

Reviewer #3: Yes

3. Have the authors made all data underlying the findings in their manuscript fully available?

Reviewer #1: Yes

Reviewer #2: Yes

Reviewer #3: Yes

4. Is the manuscript presented in an intelligible fashion and written in standard English?

Reviewer #1: Yes

Reviewer #2: No

Reviewer #3: Yes

5. Review Comments to the Author

Reviewer #1: Review

Estimates of functional muscle strength from a novel progressive lateral step-up test

are feasible, reliable, and related to physical activity in children with cerebral palsy

I thank the authors for this very interesting paper. It will be very relevant for the clinical field.

Some questions arose during the reading of the paper.

I really miss the ICF-CY as a model to use for an explanation of the relevance of the outcome and its use in the clinical field.

I miss more actual references as aertssen. w to use in the discussion.

I do not understand the reasons why 10-15-20 cm of step height are chosen and why LSUT is used instead of FSUT. What is the relation between these heights and functional environment ( stairs, street ramps…….)

As a functional test, the FSUT is more feasible related to its functional value – stairclimbing-

The height of the steps will have relevance but should be related to leg length and especially tibia length.

The explanation of the composite score

A composite LSUT score was generated by multiplying repetitions at 10 cm by 1, repetitions at 15 cm by 1.5, and repetitions at 20 cm by 2, then adding the total score.

I miss a reference and do not understand the multiplying score.

Reliability

Please relate to the cosmin guideline.

Reliability is assessed in 20 children. That is a really low number and the outcome is not yet really transferable according to the cosmin. Please add this in the discussion

Explain in the method the value of SEM and SDC and compare with the studies of Verschuren en aertssen of it is in the same line.

Please go in the discussion in more details in on the MIC and SDC and what is seen in other studies about the changes due to interventions in CP and if we can use the LSUT for evaluation for interventions

I hope you can address these questions and make the paper more valuable for the clinical field

best

Reviewer #2: Dear Author,

I have opportunity to review the manuscript titled " Estimates of functional muscle strength from a novel progressive lateral step-up test are feasible, reliable, and related to physical activity in children with cerebral palsy".

Individuals with CP experience impairments of body structures and functions, such as in strength, spasticity, ROM, co-contractions, low motor control, kinaesthetic ability, dyscoordination which, in turn, may limit their functional mobility. The relationship among these impairments is still under investigation with researchers focusing mainly upon the effect of strength, spasticity and ROM on the functional mobility of individuals with CP. A variety of measures have been used by researchers in the past to assess functional mobility of individuals with CP. The most common is the gross motor function measure (GMFM) which has been widely used and may be perceived as the golden standard. The LSUT may be perceived as a valid instrument for assessing the functional mobility of adolescents with CP. The LSUT exhibited sufficient concurrent and construct validity evidence and supported the sample specific validity evidence theory. In the light of this information, I carefully read the purpose and methodology of the article. My comments for the study are below:

- The first thing that caught my attention while reading the study was the study schedule. The study data were collected over a period of more than 10 years and time intervals. Factors that may be present in this process (such as the assessor, the environment in which the assessment is made) may affect the results. This unfortunately reduces confidence in the results.

- There is confusion about the number of participants in the article. The study included 45 children with CP, why only 20 children had LSUT scores calculated.

- There is validity evidence of the LSUT for adolescents with spastic cerebral palsy (DOI: 10.3109/09638288.2012.711896). I am not sure what the development of a new method by changing the step heights would contribute to the literature. I think that the purpose and design of the study will not provide an adequate contribution to the literature.

Reviewer #3: Review comments on Manuscript Number: PONE-D-23-23848. Entitled “Estimates of functional muscle strength from a novel progressive lateral step-up test are feasible, reliable, and related to physical activity in children with cerebral palsy"

Overall, the idea of research is very interesting to be studied nowadays and paper is coherently developed. However, there are some comments and suggestions.

Title

- Well structured

Abstract

- It is recommended to write 5 to 7 keywords in alphabetical order.

Introduction

- Well structured

Subjects and methods

- The authors recruited ambulant children with CP. However, there is great difference between unilateral and bilateral CP cases regarding energy expenditure, balance , strength. Authors should clarify which subtype of CP was included in their study.

Statistical analysis

- Well structured

Discussion

- Well structured

References

Update your References

6. PLOS authors have the option to publish the peer review history of their article (what does this mean?). If published, this will include your full peer review and any attached files.

Reviewer #1: **Yes: **e.a.a. rameckers

Reviewer #2: **Yes: **Ismail Ozsoy

Reviewer #3: No

---

## [Author Response · Author response to Decision Letter 0]

7 Jun 2024

We appreciate the thorough review of our manuscript and thoughtful feedback. We have addressed all journal and reviewer concerns and provide a summary below. We have included a manuscript file that has changes highlighted in yellow. We also include a clean manuscript file. We believe the manuscript if improved substantially and hope it is now acceptable for publication.

Journal Requirements:

Comment 1. Please ensure that your manuscript meets PLOS ONE's style requirements, including those for file naming. The PLOS ONE style templates can be found at

Response: We have ensured that the manuscript meets PLOS One’s style requirements. (Trevor will make sure to name files correctly during submission)

Comment 2. Please provide additional details regarding participant consent. In the ethics statement in the Methods and online submission information, please ensure that you have specified (1) whether consent was informed and (2) what type you obtained (for instance, written or verbal, and if verbal, how it was documented and witnessed). If your study included minors, state whether you obtained consent from parents or guardians. If the need for consent was waived by the ethics committee, please include this information.

Response: We have provided additional details about the consenting process (Ln 113-118). 

Response: We do not have any supporting files. We have updated in-text citations.

Reviewer #1: Review

Estimates of functional muscle strength from a novel progressive lateral step-up test

are feasible, reliable, and related to physical activity in children with cerebral palsy (CP).

Comment 1: I thank the authors for this very interesting paper. It will be very relevant for the clinical field. Some questions arose during the reading of the paper. 

Response: Thank you.

Comment 2: I really miss the ICF-CY as a model to use for an explanation of the relevance of the outcome and its use in the clinical field.

-I miss more actual references as aertssen. w to use in the discussion.

Response: Integration of ICF core sets for children with CP has been added in the introduction (ln 67-73) and discussion (382-386). Aertssen et al. 2019 is added to the references [23] and included in the discussion (Ln 419-428 and 435-437).

Comment 3: I do not understand the reasons why 10-15-20 cm of step height are chosen and why LSUT is used instead of FSUT. What is the relation between these heights and functional environment (stairs, street ramps…….)

Response: A maximum residential stair step height as outlined by the International Codes Council is ~ 20 cm [30] and the LSUT with that approximate step height (20-21 cm) has been previously used and validated as a test of physical function in ambulatory children and adolescents with CP [21, 23, 35]. The lower heights (10 cm and 15 cm) were included to provide greater accessibility to children with CP with a wider range of functional capabilities. Moreover, the most common curb height in residential areas in the U.S. is approximately 15 cm [31]. LSUT was chosen over FSUT due to use of LSUT in previous studies, where it was found to be both related to motor function and mobility [reference 20-23] and utilized as a lower extremity strength-building exercise in children with CP [25]. It has also been used to demonstrate suppressed activation of the pre-frontal cortex in children with CP during a functional muscle strength test [32].

Comment 4: As a functional test, the FSUT is more feasible related to its functional value – stairclimbing-

The height of the steps will have relevance but should be related to leg length and especially tibia length.

Response: The intention was to utilize a test aimed at representing a child’s functional muscle strength along with mobility/stability of the lower extremities, and their capacity to participate in physical activity and tasks of daily living, not stair-stepping alone. Also, as discussed in our response to Comment 3, the LSUT has been previously utilized as a measurement of physical function in children with CP [21, 23,25]. We agree that the height of the step has relevance. Adjusting the height of the step based on height, knee height, or tibia length is one way to account for size differences. Alternatively, having norms based on height, knee height, tibia length, and/or age may also improve the ability to assess functional muscle strength in children with CP, which we indicate in the Discussion (Ln 403-405).

Comment 5: The explanation of the composite score

A composite LSUT score was generated by multiplying repetitions at 10 cm by 1, repetitions at 15 cm by 1.5, and repetitions at 20 cm by 2, then adding the total score.

I miss a reference and do not understand the multiplying score. 

Response: An increase in step height represents an increase in difficulty, thus we wanted this to be reflected in the composite score. We did this by multiplying the repetitions completed at each step by a factor consistent with the increase in step height (i.e. 1 for 10 cm, 1.5 for 15 cm, and 2 for 20 cm). For example, for a participant who completed 10 steps at each height, their composite score would be calculated as follows: 10(1) + 10 (1.5) + 10 (2) = 45. We now include the reference and the following sentence in the Methods section (ln 168-170): ” A version of the LSUT and the composite score have been used to assess functional muscle strength and its relationship with pre-frontal cortex activity in children with CP [32]. “ 

Comment 6: Reliability - Please relate to the cosmin guideline.

Reliability is assessed in 20 children. That is a really low number and the outcome is not yet really transferable according to the cosmin. Please add this in the discussion. 

Response: We added information to the discussion that aligns with COSMIN guidelines for minimizing measurement bias (Ln 503-514). The COSMIN guidelines do not have specific guidance on sample size for tests with expected intraclass correlations (ICC) > 0.8. However, minimum sample sizes for ICCs of 0.6, 0.7, and 0.8 when the default range of 0.3 is used are 100, 50 and 30, respectively. Hence, considering the high ICCs for test-retest of the LSUT at each individual step height and the composite score (ICC range = 0.914 to 0.965), the sample size of 20 used in the present study seems appropriate. Furthermore, it has been recommended to minimize the amount of testing done to estimate ICC and SEM to alleviate unnecessary burden on research participants [41]. This is especially important for populations like CP because they are often burdened by their clinical care. We have also included Walter et al. (1998) as justification for the current sample size, as the sample size calculation outlined suggests that 20 participants is an adequate sample size to discriminate between good (ICC > 0.7) and excellent reliability (ICC > 0.9) [42].

Comment 7: Reliability - Explain in the method the value of SEM and SDC and compare with the studies of Verschuren en aertssen of it is in the same line.

Response: A better explanation of the standard error of measurement (SEM) and minimal detectable change (MDC), as well as the minimal important difference (MID) has been provided in the methods section (Ln 225-240). A comparison to the studies of Verschuren and Aertessen is provided in the discussion (Ln 419-428).

Comment 8: Reliability - Please go in the discussion in more details in on the MIC and SDC and what is seen in other studies about the changes due to interventions in CP and if we can use the LSUT for evaluation for interventions. 

Response: An addition has been made to the limitations discussing the size of the SDC relative to the magnitude of change after intervention in children with CP (Ln 503-524), as well as the use of MID (Ln 520-524).

I hope you can address these questions and make the paper more valuable for the clinical field

best

Reviewer #2: Dear Author, 

Individuals with CP experience impairments of body structures and functions, such as in strength, spasticity, ROM, co-contractions, low motor control, kinesthetic ability, dyscoordination which, in turn, may limit their functional mobility. The relationship among these impairments is still under investigation with researchers focusing mainly upon the effect of strength, spasticity and ROM on the functional mobility of individuals with CP.

A variety of measures have been used by researchers in the past to assess functional mobility of individuals with CP. The most common is the gross motor function measure (GMFM) which has been widely used and may be perceived as the golden standard. The LSUT may be perceived as a valid instrument for assessing the functional mobility of adolescents with CP. The LSUT exhibited sufficient concurrent and construct validity evidence and supported the sample specific validity evidence theory. In the light of this information, I carefully read the purpose and methodology of the article. My comments for the study are below:

Comment 1: The first thing that caught my attention while reading the study was the study schedule. The study data were collected over a period of more than 10 years and time intervals. Factors that may be present in this process (such as the assessor, the environment in which the assessment is made) may affect the results. This unfortunately reduces confidence in the results.

Response: Despite the long data collection period, the study followed a standard protocol that was co-developed and overseen by the senior author (Modlesky). We added this limitation to the discussion (Ln 515-516).

Comment 2: There is confusion about the number of participants in the article. The study included 45 children with CP, why only 20 children had LSUT scores calculated.

Response: 45 children with CP and 45 typically developing children participated in the cross-sectional aspect of the study in which we determined the relationship between functional muscle strength assessed using the LSUT and physical activity. 20 of the 45 children with CP returned for a follow up visit 1 month later to assess the reliability of the LSUT. We clarify this in the Methods (ln 95-99).

Comment 3: There is validity evidence of the LSUT for adolescents with spastic cerebral palsy (DOI: 10.3109/09638288.2012.711896). I am not sure what the development of a new method by changing the step heights would contribute to the literature. I think that the purpose and design of the study will not provide an adequate contribution to the literature.

Response: The contribution and benefits of the progressive LSUT developed in this study are outlined in the introduction (ln 73-80) and discussion (ln 360-372 and 392-396) (see below). Briefly, functional tasks in the real-world setting require the navigation of multiple step heights. Most previous studies only used one step height. Furthermore, by using multiple step heights, you can evaluate the lower body functional muscle strength in a larger number of children with CP and a wider functional range. In the present study, only 69 % of the children could complete the test at the 20 cm step height without assistance. Whereas, 87 % of the children were able to complete the test at the 10 cm step height without assistance.

Lines 73-80: “However, one limitation of the LSUT employed in previous studies is the use of a single step height [20-22]. The functional task of stepping in real-world settings requires individuals with CP to adapt based on variable conditions, such as the height of a step….”

Line 360-372: The present study demonstrates that a progressive LSUT including multiple step heights enables the evaluation of lower-body functional muscle strength in a larger number of children with CP than an LSUT that includes a single step height.”

Lines 392-396: The importance of using multiple step heights to assess functional muscle strength was demonstrated by the observation that only 69 % of the children could complete any unassisted repetitions at the 15 or 20 cm step heights, which are in the range of typical step heights used as part of a LSUT [20-23]. On the other hand, 87 % were able to complete the LSUT at the 10 cm step height without assistance. Thus, including a step height of 10 cm as part of the LSUT allowed for an evaluation that spanned a wider range of functional muscle strength.”

Moreover, the present study was the first to detect a positive relationship between lower-body functional muscle strength and physical activity in children with CP (Ln, 58-62 and Ln 376-381) (see below).

Lines 58-62: “It is plausible that the limited relationship between muscle strength and physical activity in children with CP is due to the methodology employed to assess muscle strength. Tests that assess functional muscle strength and involve the coordination of multiple joints to move the body through open space may be stronger predictors of physical activity than traditional muscle strength tests focused on isolated joint movements [18].”

Lines 376-381: “Moreover, the present study is the first to demonstrate a positive relationship between lower-body functional muscle strength, as assessed using a novel progressive LSUT, and physical activity in ambulatory children with CP. This unique observation suggests that lower-body functional muscle strength is an important factor in the physical activity of ambulatory children with CP and the methods used to assess them should be considered when evaluating their relationship.”

Reviewer #3: Review comments on Manuscript Number: PONE-D-23-23848. Entitled “Estimates of functional muscle strength from a novel progressive lateral step-up test are feasible, reliable, and related to physical activity in children with cerebral palsy"

Overall, the idea of research is very interesting to be studied nowadays and paper is coherently developed. However, there are some comments and suggestions.

Title

- Well structured

Response: Thank you.

Abstract

- It is recommended to write 5 to 7 keywords in alphabetical order. 

Response: Done.

Introduction

- Well structured

Response: Thank you.

Subjects and methods

- The authors recruited ambulant children with CP. However, there is great difference between unilateral and bilateral CP cases regarding energy expenditure, balance, strength. Authors should clarify which subtype of CP was included in their study. 

Response: Unilateral and bilateral children with CP were included in the study. This information is now included in Table 1. When we included CP type in the regression analysis, it did not have a significant effect on the relationships between LSUT performance and ankle physical activity. However, the relationships between LSUT performance and physical activity at the hip were no longer statistically significant (Ln 325-327). Type was also added to the multiple regression analysis (Table 2), where it was not a statistically significant contributor to any model.

Statistical analysis

- Well structured

Response: Thank you.

Discussion

- Well structured

Response: Thank you.

References

Update your References. 

Response: Done.

---

## [Decision Letter · Decision Letter 1]

20 Jun 2024

Estimates of functional muscle strength from a novel progressive lateral step-up test are feasible, reliable, and related to physical activity in children with cerebral palsy

PONE-D-23-23848R1

Dear Dr. Modlesky,

We’re pleased to inform you that your manuscript has been judged scientifically suitable for publication and will be formally accepted for publication once it meets all outstanding technical requirements.

Kind regards,

Renato S. Melo, PhD

Academic Editor

PLOS ONE

Additional Editor Comments (optional):

Reviewers' comments:

Reviewer's Responses to Questions

**Comments to the Author**

1. If the authors have adequately addressed your comments raised in a previous round of review and you feel that this manuscript is now acceptable for publication, you may indicate that here to bypass the “Comments to the Author” section, enter your conflict of interest statement in the “Confidential to Editor” section, and submit your "Accept" recommendation.

Reviewer #3: All comments have been addressed

2. Is the manuscript technically sound, and do the data support the conclusions?

Reviewer #3: Yes

3. Has the statistical analysis been performed appropriately and rigorously? 

Reviewer #3: Yes

4. Have the authors made all data underlying the findings in their manuscript fully available?

Reviewer #3: Yes

5. Is the manuscript presented in an intelligible fashion and written in standard English?

Reviewer #3: Yes

6. Review Comments to the Author

Reviewer #3: Review comments on Manuscript Number: (PONE-D-23-23848R1) entitled ‘’ Review comments on Manuscript Number: PONE-D-23-23848. Entitled “Estimates of functional muscle strength from a novel progressive lateral step-up test are feasible, reliable, and related to physical activity in children with cerebral palsy".

Overall, this study provides a novel approach. The idea of research is very interesting, well written and reasonable. I would like to thank the authors for their successful work to address the reviewers' comments. The authors have done great efforts to accomplish this work. They fulfilled all comments and made necessary changes throughput the manuscript. I recommend accepting the manuscript its revised form.

7. PLOS authors have the option to publish the peer review history of their article (what does this mean?). If published, this will include your full peer review and any attached files.

Reviewer #3: No

---

## [Editor Report · Acceptance letter]

1 Jul 2024

PONE-D-23-23848R1 

PLOS ONE

Dear Dr. Modlesky, 

I'm pleased to inform you that your manuscript has been deemed suitable for publication in PLOS ONE. Congratulations! Your manuscript is now being handed over to our production team.

Kind regards, 

on behalf of

Dr. Renato S. Melo 

Academic Editor

PLOS ONE